# SARS-CoV-2 Associated Immune Dysregulation and COVID-Associated Pulmonary Aspergilliosis (CAPA): A Cautionary Tale

**DOI:** 10.3390/ijms23063228

**Published:** 2022-03-17

**Authors:** Dominic Adam Worku

**Affiliations:** 1Infectious Diseases and Microbiology Department, Morriston Hospital, Swansea University Health Board, Swansea SA6 6NL, UK; dominicworku@hotmail.co.uk; 2Public Health Wales, Cardiff CF10 4BZ, UK

**Keywords:** COVID, SARS-CoV-2, fungal, aspergillosis, CAPA

## Abstract

As the global SARS-CoV-2 pandemic continues to plague healthcare systems, it has become clear that opportunistic pathogens cause a considerable proportion of SARS-CoV-2-associated mortality and morbidity cases. Of these, Covid-Associated Pulmonary Aspergilliosis (CAPA) is a major concern with evidence that it occurs in the absence of traditional risk factors such as neutropenia and is diagnostically challenging for the attending physician. In this review, we focus on the immunopathology of SARS-CoV-2 and how this potentiates CAPA through dysregulation of local and systemic immunity as well as the unintended consequences of approved COVID treatments including corticosteroids and IL-6 inhibitors. Finally, we will consider how knowledge of the above may aid in the diagnosis of CAPA using current diagnostics and what treatment should be instituted in probable and confirmed cases.

## 1. Introduction

Coronaviruses are a family of enveloped, positive-sense single-stranded RNA viruses known to cause upper respiratory tract infections and have been responsible for three pandemics to date [1]. In December 2019, a novel coronavirus, Severe Acute Respiratory Syndrome Coronavirus 2 (SARS-CoV-2) or COVID-19, was identified, which as of 13 February 2022, has resulted in >400 million cases and >five million deaths worldwide [2]. Key to the pathogenicity of SARS-CoV-2 is extensive lung disease, which is due to multiple mechanisms including immune cell infiltration, alveolar capillary damage, alveolar oedema and haemorrhage alongside microthrombi formation leading to immunothrombosis. Importantly, SARS-CoV-2 has tropism for several cells outside of the respiratory system and as such can present with additional features including diarrhoea, headache and acute kidney injury [3,4]. Common laboratory findings during active disease include lymphopenia, which is proportional to the severity of disease, hyperfibrinogenaemia, hyperferritinaemia and increased d-dimer values and C-reactive protein (CRP). While 80% of COVID-19 infected patients will suffer a mild disease, 15% suffer severe disease requiring hospitalisation and 5% require ITU-level support [5,6].

Co-infection and superinfection rates in SARS-CoV-2 patients have been reported at 19% and 24%, respectively, which contribute disproportionately to recorded mortality and can be bacterial, viral or fungal in nature. While challenging to diagnose, clinical suspicion should be raised of coinfection/superinfection where there is progression of lung infiltrates, low and worsening PaO_2_/FiO_2_ ratio and majorly the persistence of fever, which is the most predictive [7,8]. Of these, COVID-19-Associated Pulmonary Aspergilliosis (CAPA) is of particular concern. CAPA affects up to 30% (3.8–35%) of the most severely affected patients who are treated in the Intensive Treatment Unit (ITU) due to the associated multiorgan failure and immunopathological changes severe disease provides and is associated with poor outcomes. Importantly, the incidence of CAPA varies considerably among countries and units and is likely underdiagnosed, further contributing to the poor outcomes observed [9,10,11,12]. While this is a new entity, it is important to recognise a similar presentation is well described in post influenza patients [13]. Both of these entities are associated with increased mortality, morbidity, duration of hospital stay and mechanical ventilation duration [14,15].

Aspergillus is a ubiquitous environmental thermotolerant fungal family consisting of >200 known species and is found primarily in the soil. Aspergillus spores can easily be aerosolized and can travel a considerable distance, allowing them be taken up by the respiratory system of humans. The average adult inhales >100 aspergillus conidia daily, which can be a coloniser of the respiratory system on diagnostic sampling. Due to their small size (2–3 µM), they can easily enter the alveolar compartments of the lungs where they can adhere to the basal lamina of airway epithelia but are normally effectively cleared in an immunocompetent host. Of all species of aspergillus, *Aspergillus fumigatus* (*A. fumigatus*) is by far the most common species isolated in both invasive fungal disease and cases of confirmed CAPA worldwide although other species have also been described [16,17].

The primary risk factor for invasive fungal disease is being immunocompromised and chiefly those who are neutropenic, which can be qualitative or quantitative in nature, highlighting the role of neutrophils in preventing fungal disease. Other risk factors include impaired physical barriers through trauma or injury as well receipt of glucocorticoids (Table 1). As such, at-risk groups include oncology, solid organ transplant recipients and haematological patients, intensive care unit populations and those with chronic and structural lung disease, as these allow a niche for inhaled aspergillus conidia to invade locally and germinate, allowing for dissemination [18]. A key difficulty, however, as highlighted is distinguishing between Aspergillus colonisation and invasive infection given the ubiquity of these fungi in clinical samples such as throat swab, tracheal aspirate, etc. Current diagnostic algorithms for Invasive Fungal Infection (IFI) including CAPA struggle in this regard although a newer diagnostic guideline (AspICU) algorithm categorises colonisation separately but at the expense of fungal biomarkers [19]. Therefore, CAPA requires strong clinical suspicion and the integration of clinical, radiological and biomarker correlates to correctly make the diagnosis, all of which individually can be difficult to intepret and may be confounded by comorbidities and implemented therapies. This assessment therefore in severely affected SARS-CoV-2 patients requires a plethora of investigations and expertise to form a multidisciplinary team including radiologists, intensivists, infectious disease experts and microbiologists. Complicating this further is the lack of a unifying definition of CAPA and the lack of standardised treatment regimes, which are based on invasive asperigllosis and not aspergillosis, complicating SARS-CoV-2 infection [9,10].

## 2. SARS-CoV-2 and the Immune System

COVID-19 is caused by the SARS-CoV-2 virus, of which several variants have been identified to date including most recently the Omicron variant (November 2021) [20]. While SARS-CoV-2 has a very similar structure to the related coronaviruses, Middle Eastern Respiratory Syndrome (MERS) (50% homology) and Severe Acute Respiratory Syndrome (SARS) (79% homology), which have both resulted in pandemics, there are important differences that have led to its global dissemination [1]. Common among the coronaviruses are four conserved structural proteins, Spike (S), Envelope (E), Nucleocapsid (N) and Envelope (E), alongside which there are associated non-structural proteins (NSPs) and accessory proteins. Of these, the (S) protein has been found to be important in SARS-CoV-2 evasion of humoral immune responses while NSPs have been found to modulate and suppress interferon expression, which is a key facet of antiviral immunity, thereby enhancing virulence [21].

Unlike in MERS and SARS, SARS-CoV-2 has multiple changes in the (S) protein, which is implicated in viral envelope cell binding and subsequent viral entry. The Angiotensin Converting Enzyme 2 (ACE2) receptor, a component of the renin and angiotensin system, is the primary receptor for SARS-CoV and SARS-CoV-2 infection and is highly expressed on type 2 pnuemocytes and epithelial cells of the intestine and kidney, helping to explain the predilection for extrapulmonary symptoms such as diarrhoea and acute kidney injury in infected individuals. For effective SARS-CoV-2 cellular infection, transmembrane serine protease 2 (TMPRSS2), a cell surface protease, is required, which cleaves and activates the S1 and S2 subunits of the (S) protein. [22,23]. More recently, it is has been appreciated that there is also expression of the ACE2 receptor on select CD68+ macrophages, suggesting there is direct infection of cells of the innate immune system allowing for immune evasion and helping to explain the high incidence of macrophage activation syndrome and the cytokine storm noted with resultant complications in those with severe disease [24].

Upon SARS-CoV-2 entry into the cell, there is activation of the cellular immune response through the activation of pathogen recognition receptors (PRRs) that recognise Pathogen-Associated Molecular Patterns (PAMPs), which may be bacterial, viral or fungal in nature. Several PRR subtypes exist including Toll-like Receptors (TLRs), C-type Lectins (CTLs), Retinoic Acid Inducible I like receptors (RIG-I) and Nucleotide-binding oligomerization domain Receptors (NODs). Together these sense multiple different types of pathogens and substrate types. TLR 3, 7 and 8 recognise SARS-CoV-2 RNA in different cellular compartments including the endosome while TLR4 can directly recognise the (S) protein, leading to downstream effects including myeloid differentiation primary response 88 (MyD88) mediated signalling that leads to nuclear factor-*κB* (NF-κB) activation stimulating the host inflammatory response [25,26]. Interestingly this interaction between TLR4 and the (S) protein has been found to reciprocally increase ACE2 receptor expression, which has been hypothesised to lead to greater cellular infection and hyperinflammation [27]. Other significant PRR-PAMP interactions include SARS-CoV-2 RNA and Melanoma Differentiation-associated protein 5 (MDA5), a RIG-I like receptor and RNA helicase enzyme [21]. Upon this interaction, enhanced Mitochondrial Antiviral Signalling (MAVS) occurs, leading to further activation of NF-κB through IκB phosphorylation and proinflammatory cytokine release, thereby stimulating and localising neutrophils and macrophages as well as phosphorylating Interferon Regulatory Factor 3 and 7 (IRF 3, IRF7), which can then bind to the promoter region of interferon β (INFβ [28]. Through the above, we have upregulation of cellular and systemic anti-viral immunity and stimulated IFNβ, which interacts with the IFNα/β receptor in an auto/paracrine fashion leading to local response. IFNβ release leads to activation of Janus Kinase 1 (JAK1) and the tyrosine kinase TYK 2 leading to STAT1 signalling (signal transducer and activator of transcription 1). STAT1, STAT2 and IRF9 then as a complex (ISGF3) bind onto DNA and lead to promotor binding and interaction at the interferon-stimulated response element (ISRE) with subsequent interferon-stimulated gene (ISG) activation [21]. The interferon response involves >300 molecules and the transition of the cell into an antiviral state, which includes enhanced antigen presentation as well as effector function of circulating and local B, T, Dendritic and Natural Killer Cells. One key product includes 2′-5′-oligoadenylate synthetase-RNase L that has inherent endoribonuclease activity and can lead to single-strand RNA (ssRNA) degradation while amplifying the antiviral response through positive feedback of IFNβ signalling [29,30]. 

To avoid interferon responses, multiple viruses have developed multiple methods to avoid detection and enhance their pathogenicity. SARS-CoV-2 Open reading frame 6 and 8 (ORF6 and ORF8) proteins can bind to and inhibit interferon signalling by blocking STAT1 importation into the nucleus as well as nuclear export of PRR mRNA in a similar but less efficient manner to SARS-CoV [31,32]. Moreover, there is evidence of detectable mutations in interferon signalling pathways (e.g., TLR3 and IRF7) in those with severe SARS-CoV-2 disease as well as enhanced autoantibody levels to type 1 interferons with neutralising activity in vitro in ~13% of patients with severe/life threatening SARS-CoV-2 infection. This seroprevelance is approximately 15 times greater than in healthy controls, further hampering effective antiviral immunity and helping to predispose patients to severe disease. As such, their presence could be used as a risk stratification measure in future if they can be routinely and reliably tested in the clinical environment [21,33,34].

The delays in interferon responses create time for viral replication to occur unhindered, allowing for dissemination and a hyperinflammatory response, which is mediated by cell injury and the release of danger-associated molecular patterns (DAMPs) that lead to further stimulation of the innate immune response [22]. While there is aberrant interferon responses in SARS-CoV-2, early trials investigating the use of interferons have shown SARS-CoV-2 to be intrinsically more susceptible to interferons than other coronaviruses and in select studies has been found to reduce viral shedding time, hospital stay and duration of illness; however, this has not been demonstrated in larger randomised contrail trials (RCTs). Moreover, combination therapy of interferons with existing antiviral medications such as Remdesevir, a nucleotide analogue and viral RNA-dependent RNA polymerase inhibitor, did not improve outcomes. As such, interferons are not recommended in unwell SARS-CoV-2 patients. [35,36]. Due to perturbations in antiviral immunity in the setting of SARS-CoV-2, disseminated infection can occur with simultaneous activation of multiple components of the innate and adaptive immune system. When this process occurs in an uncontrolled fashion, a deleterious and futile immune response is produced, characterized by a cytokine milieu called the ‘cytokine storm’. This process is frequently observed in patients requiring ITU support and who succumb to SARS-CoV-2 infection. It is typified by greatly increased levels of the proinflammatory cytokines IL-1α, IL-6 and TNFα, which are produced primarily by macrophages, mast cells, endothelial and epithelial cells through NF-kB induction, which is seen as the orchestrator of this clinical presentation, making it an attractive target for intelligent drug design [5]. The levels of these cytokines have been found to correlate with nasopharyngeal SARS-CoV-2 viral load, with their release into the local and systemic circulation leading to the further localisation, influx and hyperactivation of macrophages, monocytes and dendritic cells, leading to perpetual inflammation and tissue injury. However, it is appreciated that in SARS-CoV-2 infection, there are both reduced numbers and cytotoxic activity of circulating NK, CD4+ and CD8+T cells with a move towards senescence further contributing to suboptimal outcomes [5,37]. In these circumstances due to extensive and diffuse alveolar damage, adult onset respiratory distress syndrome (ARDS) and haemodynamic stability, multi-organ failure and death can quickly follow unless treatment is initiated. Importantly, this phenomenon is not unique to SARS-CoV-2 infection and may also be seen in H1N1 influenza, Epstein Barr Virus (EBV), SARS-COV and MERS, although not as frequently [38].

The complement system is an important part of the innate immunity and is involved in the detection of microorganisms and the priming of both arms of the immune system and is increasingly being recognised as an important contributor to SARS-CoV-2 pathology. While there are three methods of complement activation (classical, lectin and alternative), all mechanisms lead to the formation of several important molecules including C3b, which can opsonize pathogens for phagocytosis by circulating neutrophils and macrophages, and C5a, an anaphylatoxin and neutrophil activator, attractor and aggregator, and the membrane attack complex (C5b-C9), which leads to bacterial and infected host cell death [39]. Similar to SARS, SARS-COV-2 shows the ability to activate the lectin pathway by interacting with mannose-binding lectin (MBL). In SARS-CoV-2-infected patients, there is high-level deposition of MBL, C3, C4 and C5b-9 on pulmonary epithelial cells and pnuemocytes, leading to tissue injury. Moreover, the deposition of this complement leads to local endothelial injury and exposure of the basement membrane. This allows for subsequent C5a and C5a receptor binding, leading to endothelial cell activation, platelet granule degranulation and neutrophil extracellular trap (NET) release, which contribute to resultant immunothrombosis [24]. Neutrophils are the predominant circulating leucocyte and are raised in SARS-CoV-2 and are localised to sites of SARS-CoV-2 replication such as the pharynx and lung. One key contribution of neutrophils to SARS-CoV-2 is NETs, which are composed of multiple extracellular chromatin fibres that are covered in antibacterial proteins including cathelicidins, neutrophil elastase (NE), myeloperoxidase (MPO), proteinase 3, pentraxan 3 and α-defensins. These are extruded in a lytic fashion from the cell membrane and represent a form of cell death for the neutrophil [3,40]. NETs are released to physically limit the invasion of pathogens by entrapping bacteria, fungi, protozoa and viruses, which are too large to be phagocytosed and are released on exposure to SARS-CoV-2 in an ACE2-dependent manner and are demonstrably raised in the serum of patients with severe SARS-CoV-2 infection and highly concentrated in the pulmonary tissue of autopsy specimens. Moreover, higher serum MPO-DNA levels have been found to predict the risk of poor outcomes such as intubation (*p* < 0.001) and death (*p* < 0.0005) [3,41,42]. Overall, NETs’ pathological role in severe SARS-CoV-2 includes vaso-occlusion, endothelilits and pulmonary epithelial cell necrosis while also increasing IL-1β production, leading to further macrophage recruitment and activation. This local tissue damage exacerbates the inflammatory cascade and in doing so, contributes to the poor outcomes of severe SARS-CoV-2 infection and NET formation; NETosis is a mechanism and a potential drug target of the future [3,43,44].

## 3. Aspergillus in Health and SARS-CoV-2

In health, the first point of contact of inhaled conidia will be the airway epithelium, which is a ciliated stratified epithelium capable of eliminating conidia through secreted mucus by goblet cells; the coordinated beating of cilia and the mucociliary escalator lead to either their swallowing or expectoration [10,19,45]. Importantly however, *A. fumigatus* can circumvent these mechanisms through toxin production and in the setting of concurrent disease, can enter the lower airways. Upon reaching the lower airways, conidia will be exposed to different antimicrobial proteins including defensins and lactoferrins, which will be upregulated in the presence of *A. fumigatus*. Moreover, there is direct interaction with cells of the innate system through PAMP-PRR interaction with pulmonary macrophages able to produce chitinase, which breaks down the aspergillus cell wall while circulating neutrophils can produce NETs and pentraxan 3. Pentraxan 3 is a soluble recognition receptor also recognized by epithelial cells and can opsonise conidia in an FcyRII-dependent manner for destruction by neutrophils and macrophages and prevent their germination into more invasive hyphae. It also works to activate the complement system against aspergillus by C3b binding, leading to alternate pathway activation. Aspergillus conidia, however, have mechanisms to evade complement activation including the masking of surface (1,3)-β-d-Glucan (BDG) and mannose by RodA, a surface hydrophobin, and through recruitment of complement inhibitors such as Aspf2 as well as the degradation of C3, C4 and C5 through protease secretion [46,47]. This critical role of pentraxan in preventing invasive aspergillosis is evident in pentraxan knock-out mice who were more susceptible to disease and showed clinical improvement upon its administration. Interestingly, serum pentraxan levels in ITU SARS-CoV-2 patients are unaffected versus those with sepsis from other pathogens, which may explain the heightened risk of CAPA in these patients [47]. Dendritic cells work as a link between activating the innate and adaptive immune response while also directly phagocytosing aspergillus and leading to further increase in proinflammatory cytokine release as well as an adaptive Th1-mediated response. Indeed, both Th1 and Th17 adaptive responses predominate in fungal disease with IL-17 binding to swollen conidia versus resting conidia, which may exist in biofilms, and resulting in further clearance driven by alveolar macrophages [10,22]. Paradoxically, increased Th17 signalling in the setting of severe SARS-CoV-2 infection can lead to increased invasive aspergillosis risk [48,49].Responses in SARS-CoV-2 are attenuated with autoantibody formation occurring in ~10% of severely ill SARS-CoV-2 patients. Diminished IFN-β secretion reduces the efficiency of dendritic cell-mediated Th1 responses that are important in aspergillus eradication while decreased Type 3 interferons, e.g., IFN-λ, can reduce neutrophil activity against *A. fumigatus,* which is indispensable for the prevention of subsequent invasive disease [50].

As we have discussed, severe SARS-CoV-2 is associated with widespread immune change and also diffuse pulmonary epithelial and alveolar damage, which creates a niche for secondary infection with CAPA, complicating SARS-CoV-2 infection in 3.8–35% of ITU patients depending on the population studied [10]. Typical cases occur in ITU SARS-CoV-2 patients who are older, have a higher sequential organ failure assessment (SOFA) score, are mechanically ventilated and require vasopressor support. The mortality of 565 mechanically ventilated patients with proven/probable CAPA in the large French national multicentre MYCOVID study was 61.8% [51]. Important to note is that CAPA’s mortality persists even after adjustment for age, the SOFA score and the need for organ support against SARS-CoV-2 controls. As such, severe COVID19 is itself a risk factor for CAPA with it being an extremely rare disease in non-severe SARS-CoV-2 infection [52,53]. Typical risk factors for invasive fungal infections such as aspergillosis include underlying haematological disease and immunosuppression, which can be either primary or secondary to chiefly chemotherapy or corticosteroid therapy. More recently, ITU stay, broad-spectrum antibiotic use and comorbidities such as chronic obstructive pulmonary disease (COPD), cirrhosis, chronic kidney disease and non-haematological cancer are being recognised as additional risk factors for invasive fungal disease [54]. Severe SARS-CoV-2 infection is characterised by the cytokine storm with IL-6 critical to this process and the attenuation of antiviral immunity. IL-6 is upregulated in cells infected with aspergillus, highlighting the role that its inhibition has in the pathogenesis of CAPA [22,55]. Moreover, sustained high levels of IL-6 and TNFα can lead to reduced macrophage maturation and reduced Major histocompatibility complex (MHC) Class 2 expression, compounding the risk of IFI further [56]. Antibiotic use in SARS-CoV-2 is high and often in the absence of supporting evidence and can be a cause of considerable harm. In a recent meta-analysis of >30,000 SARS-CoV-2 patients, ~75% received antibiotics despite the bacterial co-infection rate being only 8.6%. Epidemiological evidence suggests broad-spectrum antibiotic use may increase CAPA risk through selection processes although this is not a universal finding [57,58]. A possible relationship between antibiotic use and CAPA could be through increased release of bacterial PAMPs, which upon leading to induction of the immune response, increases aspergillus gliotoxin release, which is highest in *A. fumigatus* and has broad immunosuppressive actions. These include inhibition of NF-kB leading to enhanced apoptosis, reduced reactive oxygen species (ROS) formation by phagocytes as well as decreased cytotoxic T-cell activity [59,60].

The risk from CAPA is also iatrogenic in nature and greatly increased because of the treatments we have deployed. The Randomised Evaluation of COVID-19 Therapy (RECOVERY) is an international clinical trial that has synthesised evidence on potential treatments for COVID-19 since 2020. One of the key and earliest discoveries was the beneficial outcomes found in patients who received Dexamethasone, a high-potency glucocorticoid. At present, 6 mg Dexamethasone (or alternative steroid dose equivalent) is a key pillar of COVID-19 treatment, given for up to 10 days, reducing mortality by 36% and 18% in invasively ventilated and non-invasively ventilated patients, respectively [61]. Dexamethasone and other glucocorticoids have several key side effects that include immunosuppression through the curtailing of both the adaptive and innate immune systems. Mechanisms of this include decreased phagocyte adherence to endothelium, extravasation, ROS formation and phagocytosis by inhibiting the recruitment of key proteins such as the LCSII protein. This allows for the persistence of aspergillosis conidia and their subsequent germination into hyphae. In addition, glucocorticoids can reduce NF-kB activation, leading to depressed priming of both arms of the immune system with evidence glucocorticoids can directly promote aspergillosis growth—something that was not observed with other human steroids. It is of no surprise, therefore, of the correlation between glucocorticoid use in diverse patient populations (e.g., COPD, collagen disorders, cancer) and invasive aspergillosis infection [62,63,64].

In a meta-analysis of cohort studies of ITU COVID-19 patients, glucocorticoid use was linked with a 10-fold increase in candiademia and a relative risk ratio for CAPA of 1.98 (95% CI 1.08–3.63) [65]. Other immunosuppressive treatments deployed in the SARS-CoV-2 arena include the soluble IL-6 receptor inhibitor Tocilizumab, which is used for a diverse range of rheumatological conditions. Through effective inhibition of IL-6, decreased inflammation, cytokine storm progression and tissue fibrosis are possible with evidence of reconstituted cytotoxic potential of immune cells including NK cells. Overall efficacy suggests that when administered at 8 mg/kg, Tocilizumab significantly reduces mortality (OR 0.62; CI 0.55–0.70, *p* < 0.00001), and length of hospital stay, particularly in mechanically ventilated ITU patients [65,66]. However, similar to Dexamethasone, coinfection and superinfection are key risks and may go undetected due to delayed/absent CRP rise due to IL-6’s role in CRP production. In one retrospective observational study, 26.7% of rheumatological patients post Tocilizumab suffered infectious complications at a median of 10.5 days post administration. In a meta-analysis of 33 studies in SARS-CoV-2 patients, there was no significantly increased incidence of secondary infection (OR 1.12 95% CI 0.87–1.43, *p* = 0.0376) apart from fungal infection in Tocilizumab recipients (OR 2.02, *p* = 0.036) [67,68].

The risk of CAPA, particularly with Tocilizumab, is partly explained by the importance of IL-6 in epithelial integrity with inhibition increasing the risk of invasive disease of any germinating conidia [69]. The observations for fungal infection corroborate a recent multinational multicentre trial of 592 ITU patients where the hazard ratio for CAPA in Tocilizumab recipients was 2.45 (95% CI 1.41–4.25) and was higher than that for dexamethasone (HR 1.01). Importantly, this risk was compounded in those requiring invasive ventilation; HR 3.4 (95% CI 1.84–6.25) and of older age; HR 1.18. It is important to note that a consistent observation between CAPA and Tocilizumab without concurrent corticosteroid use has not been found [56]. Because of the above mechanisms, tolerance of the lung microenvironment to Aspergillus conidia is achieved, allowing them to propagate and germinate, which is key to their subsequent angioinvasion [70]. This invasion leads to the occlusion of small- to medium-sized pulmonary arteries leading to subsequent pulmonary necrosis, infarction and haemorrhage, leading to the clinical worsening that is observed. Moreover, during this process, an embolic phenomenon may occur through haematogenous spread of hyphae to other organs. This can result in subsequent thrombosis and infarction of distant tissues and involvement of other organs with complications of invasive aspergillus including endocarditis, central nervous system infection and stroke and may eventually culminate in multiorgan failure and death [71,72].

## 4. Challenges in Diagnosing CAPA

Diagnosing CAPA is challenging and requires the correct interpretation of clinical, radiological and biochemical parameters including serum fungal biomarkers. At present, there is no standardised definition for CAPA. Of the diagnostic algorithms that exist to help in this regard with respect to IFI, there are major limitations and assignment of patients as proven, putable, possible and probable. To be defined as a proven case for histological examination, which is the gold standard test, is not always appropriate considering the coagulopathy seen in the critically unwell and the risks in performing bronchoscopy given the inherent risk of aerosol contamination of the surroundings and nosocomial SARS-CoV-2 transmission [73].

The European Organization for the Research and Treatment of Cancer/Mycosis Study Group (EORTC/MSGERC) algorithm, only validated in haematological patients, requires the presence of host risk factors that can be absent in SARS-CoV-2 patients and does not recognise aspergillus colonisation [19].

More recently, the AspICU algorithm, which is adapted to the ITU environment, has been produced and does differentiate between putative CAPA and colonisation but does not rely on fungal biomarkers. When compared to EORTC/MSGERC in 27 proven invasive pulmonary aspergillosis (IPA) patients, of which 15 had a positive serum galactomannan (GAL), 16 would be considered as putative through the AspICU algorithm. The ECCM/ISHAM criteria are perhaps the most clinically useful as they have wide range of clinical specimen types including tracheal aspirates and sputum although the risk of aspergillus contamination of these sources cannot be underemphasised [19,74]. Radiological findings are included in all diagnostic criteria of IFI. In severe SARS-CoV-2 imaging, findings include bilateral pulmonary infiltrates, ground glass opacities, nodules and frank consolidation, which means there is significant overlap with invasive pulmonary aspergillosis, making the radiological diagnosis of CAPA extremely difficult and as such a reliance on microbiological tests is unsurprising [75]. Of the fungal biomarkers used to diagnose CAPA, BDG and galactomannan (GAL) are the most commonly used, but neither test achieves the required sensitivity, specificity or predictive value to be used to solely diagnose IFI. Importantly, these are not available in all local laboratories and can be raised in a plethora of settings leading to false positive and negative results (Table 2) [76,77,78,79,80,81,82,83,84].

(1,3)-β-d-Glucan is fungal cell wall polysaccharide, which is found in most fungal cell walls with few exceptions and is sporadically released into the circulation during the fungal growth cycle. While the gold standard for diagnosing IFI is biopsy and identification of fungi in specimens on staining, sampling from bronchoalveolar lavage (BAL) also reports high yield and can also be tested for BDG and GAL positivity. However, as mentioned, bronchoscopy has inherent risks, being invasive and aerosol generating, with the positivity of fungal culture in invasive aspergillosis suboptimal at 45–60%. As such, serum BDG is a safer measurement with serial measurements available; indeed, there is evidence that early negative BDG or GAL during antifungal therapy can help predict positive outcomes [85]. A Cochrane review conducted in 2020 highlighted the considerable variation in sensitivity (27–100%) and specificity (0–100%) across all diagnostic BDG platforms in immunocompromised and critically ill patients although there was large variation in defined possible/probable cases of IFI amongst included studies [86]. As such, a negative result cannot exclude invasive fungal disease. Galactomannan (GAL) is a fungal cell wall polysaccharide and similar to BDG, is released particularly during aspergillus growth and development. Overall, the pooled sensitivity and specificity of serum BAL and GAL is 0.48–0.92 and 0.85–0.95, respectively, with positive serum GAL only found in invasive disease. However, there are estimates of only 20% CAPA patients having positive serum GAL with cases of negative GAL well described [74]. As such, BAL-GAL is the best type of diagnostic sampling. At present, an ODI cut-off of GAL of 1.0 in BAL has been proposed over the traditional 0.5, as this has the best overall sensitivity 0.75–0.86, with specificity 0.94–0.95, although these values are significantly reduced in those who are pre-emptively put on mould active antifungals [87].

Importantly, in those in whom Tracheal aspirates were performed, GAL ODI cut-off values ≥2.0 showed a high degree of concordance with BAL Aspergillus PCR and culture. As such, tracheal aspirates could be an underutilised sampling approach to diagnosing CAPA and identifying patients who may benefit from a more invasive BAL [12].

Recently, a serum GAL lateral flow assay (LFA) was developed that has already been used in haematological malignancy patients with excellent sensitivity (96.9%) and specificity (98%) achieved at a galactomannan index of 0.5 [81]. This has the obvious advantage of being non-invasive with a shorter turnaround time with results available between 15–30 min. When used in the setting of a mixed cohort of patients including 59 severe SARS-CoV-2 ITU patients at 0.5 ODI cut off, the sensitivity and specificity of the GAL LFA were 78.6% and 80.5%, respectively, with significant agreement with serum GAL (*p* < 0.001). While this study was limited in being a retrospective analysis hampered by its low number of proven cases and low number of CAPA cases, it illustrates the utility that this emerging technology may have in the clinical setting [88,89].

It is important to note that there are different forms of this technology with Aspergillus lateral flow devices (LFDs) and LFAs in circulation. A recent review compared the performance of both of these in serum in the setting of invasive aspergillosis in 101 post-allogenic hematopoietic stem cell transplantation patients. Overall, of 86 patients with proven and probable invasive aspergillosis, the aspergillus LFD was positive in only nine (10.5%), while the LFA performed only marginally better: 18 (20.9%) with false positive results up to 12.7%. The sensitivity of the LFD and LFA was 40% with specificity of 86.8% and 89%, respectively. The sensitivity did not improve with serial testing. Therefore, it appears the serum aspergillus LFA is slightly better as a potential screening test for allowing further investigations for IFI to be performed, including BAL, where the LFA had better performance as a diagnostic tool (sensitivity 100%, specificity 81%) [90].

Chronic Pulmonary Aspergillosis refers to a spectrum of conditions with chronic cavitary pulmonary aspergillosis (CCPA) the most common and is classically not associated with background immunosuppression. In this setting, the sensitivity and specificity of GAL in BAL were 77.2% and 77%, respectively, versus 66.7% and 63.5% in serum (cut-off index 0.4) [91]. When utilised here, the aspergillus LFD had a comparable sensitivity of 62% and 67.7%, which was broadly unchanged upon using BAL samples (sensitivity 66.7% and specificity 69.2%), which was only marginally improved when using a GAL cut-off index of 0.6 (sensitivity 72.7%, specificity 83.1%). Therefore, while the reliability of the device is insufficient, its turnaround time may be of use particularly in resource-limited countries or centres for aspergillosis screening [92]. In a retrospective testing of 238 SARS-CoV-2 patients of which 148 serum and 196 samples were obtained, the aspergillus LFA in BAL showed that at the 0.5 cut-off overall sensitivity for proven, probable and possible diagnoses of CAPA was 72% and 100% in BAL fluid and tracheal aspiration, respectively, with a corresponding specificity of 79% and 44%. In serum, however, an overall sensitivity and specificity of 20% and 93% was achieved. This highlights once again that the use of LFA over LFD for tracheal aspirate samples may assist in identifying patients who may benefit from further invasive investigations, highlighting the possible risk of CAPA early [93].

## 5. Therapeutics in CAPA

Upon diagnosis of CAPA, early treatment is important to improve mortality, which can be considerable. Voriconazole and Isavuconazole are widely used as a first-line treatment with equivalent efficacy for CAPA although with the former there are important considerations concerning drug–drug interaction through its CYP2C19 and CYP3A inhibition and therefore the need for therapeutic drug monitoring. For vorincazole, trough concentrations of 2–6 mg/L are recommended, although this may be challenging to achieve. This is particularly important in patients who may be receiving Extracorporeal Membrane Oxygenation (ECMO) as their respiratory support in whom antifungal pharmacokinetics can be severely deranged [74,94].

The preferential use of triazoles in this setting mirrors the landmark study by Herbrecht et al. 2002, which compared voriconazole and amphotericin B (AmB) in invasive aspergillosis where the mortality hazard ratio was 0.59 in the voriconazole group [95]. There are, however, no direct RCTs comparing antifungals in CAPA. Voriconazole requires therapeutic drug monitoring and can cause renal dysfunction. This is particularly the case with intravenous Voriconazole, which contains sulphobutylether-β-cyclodextrin (SBECD) that acts to improve voriconazole solubility and can accumulate in the kidneys. As highlighted, SARS-CoV-2 can cause renal dysfunction and as such, extra care must be taken to monitor for this. Moreover, the Food and Drug Agency states intravenous voriconazole use should be avoided in patients with CrCL < 50 mL/min or requiring haemodialysis. The British National Formulary does not share this view, however, and advises caution and highlights the oral route of therapy as an alternative [96,97]. In one retrospective observational study of 166 patients receiving voriconazole for at least three days, 42 patients (CrCl < 50 mL/min) received intravenous voriconazole, 77 (CrCl > 50 mL/min) received intravenous voriconazole and 47 (CrCl < 50 mL/min) received oral voriconazole. On days 3 and 7 and at the end of treatment (EOT), renal review changes as per the RIFLE criteria were found in 19 (11.4%),14 (8.4%) and 28 (16.9%) patients, respectively. In multivariate analysis, it was shown those who developed day 3 injury with ≤7 days voriconazole exposure were more likely to be on concurrent antibiotics, particularly penicillins and fluoroquinolones (OR 30.7, *p* < 0.0001), received treatment with immunosuppressive agents (OR 7.90, *p* = 0.009) and had concurrent haematological malignancy and previous fluconazole use within 30 days (OR 7.80, *p* = 0.001). However, after receiving three days of treatment, even at day 7 or EOT assessment, there was no link between the route of administration and baseline renal function and subsequent renal dysfunction. Upon reviewing overall risk factors, those who had baseline liver dysfunction had the highest risk of renal impairment with voriconazole levels ≥5.0 mcg/mL significantly associated with worsening renal function at the end of treatment [96]. This suggests IV Voriconazole is safe but needs to be considered on a case-by-case basis, particularly in the context of other employed treatments such as antimicrobials that are commonly deployed in the treatment of SARS-CoV-2. Other antifungals including Liposomal AmB are well recognised to cause significant renal toxicity but may have a role in CAPA, particularly if there is local epidemiological evidence of high triazole resistance amongst aspergillosis infections (>5% isolates), and susceptibility testing should be routinely performed. Classical mutations include the TR34/L98H mutation, which is an environmental resistance mutation and likely linked to fungicide use and has been reported globally. Triazole resistance in one case series was calculated at ~3.7% of CAPA cases in a recently published German ITU study and as such is not insignificant [98,99,100]. Therefore, clinicians should be aware of this when assessing treatment response in CAPA. At present, the duration of therapy is undefined, but most authorities agree it should be 4–6 weeks with clinical review at the end of treatment. Some experts suggest 6–12 weeks therapy with follow-up CT imaging to demonstrate resolution of lesions with longer courses emphasised in immunocompromised individuals and those with cavitatory disease [74]. Due to the above, the role of antifungal prophylaxis is being considered although this is not a practice in the setting of the related influenza-associated aspergillosis. Two observational studies that used intravenous posaconazole or inhaled amphotericin B as antifungal prophylaxis in unwell SARS-CoV-2 patients reported significantly reduced incidence of CAPA and aspergillus colonization rates; however, there was no difference in 30-day mortality versus controls [101,102]. These findings require further RCT analysis to define the role that antifungal prophylaxis may have in preventing CAPA.

In summary, given the high morbidity and mortality of CAPA, which occurs in the most severely affected SARS-CoV-2 patients and leads to progression of symptoms, during initial assessment and treatment, these cases should be under inpatient care. This should ideally include a wide array of medical specialties including Radiology, Microbiology, Infectious Diseases and Respiratory. Upon recovery, treatment should be continued until symptoms and radiological improvement are noted and upon discharge, regular outpatient follow-up is mandated.

## 6. Conclusions

While SARS-CoV-2 infection for the majority of those affected is a mild illness, in at-risk populations, it can cause severe disease leading to mechanical ventilation, haemodynamic instability and death. This novel coronavirus leads to multiple downstream immune system changes of particular antiviral and innate immunity, which allows for SARS-CoV-2 propagation and progressive lung damage and can predispose patients to the cytokine storm, particularly in those who are comorbid. Among co-infections, CAPA is an increasing concern and is a significant risk in patients who receive Tocilizumab therapy and likely remains undiagnosed in a significant proportion of infected individuals. While serum fungal biomarkers can be used, they have significant limitations, with BAL sampling achieving the highest sensitivity and specificity to identify probable cases, although there are significant risks in this arena and as such, galactomannan LFA and tracheal aspirate sampling may prove useful. Clear challenges therefore exist in improving the identification of cases and further understanding the pathophysiology of CAPA, which remains poorly understood. Treatment should be for at minimum 4–6 weeks with voriconazole with close clinical correlation and repeat imaging at the end of treatment to demonstrate cure.

## Figures and Tables

**Table 1 ijms-23-03228-t001:** Risk Factors for Invasive Fungal Disease [8,13,19].

Glucocorticoid treatment Qualitative or quantitative neutrophil deficiency (Absolute count < 0.5 × 10^9^/L Impaired integrity of physical barriers, e.g., Burns, Mucositis, Peripheral line insertion Increase gut wall permeability e.g., Surgery, Total Parenteral nutrition, Intestinal perforation Stem Cell and Solid Organ Transplantation Fungal Colonisation

**Table 2 ijms-23-03228-t002:** False positive and false negative testing in serum fungal biomarkers [76,77,78,79,80,81,82,83,84].

Biomarker	False Positive	False Negative
**β-D-Glucan (BDG)**	Germinoma Penicillin G and other antimicrobials Bacteraemia Nocardiosis IVIg Albumin Haemodialysis Glucan-containing Gauze	Lipaemic blood samples Haemolysed samples Non-BDG containing fungi (e.g., Cryptococcus, Mucormycosis, Blastomycosis)
**Galactomannan (GAL)**	Piperacillin tazobactam Co-amoxiclav Ampicillin Phenoxymethylpenicillin Cefepime IVIg TPN Increasing age or newborn Antifungal therapy Cotton gauze Damaged intestinal mucosa	Variable GAL release Renal dialysis

## Data Availability

Not applicable.

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
