# Peer review of "SARS-CoV-2 Associated Immune Dysregulation and COVID-Associated Pulmonary Aspergilliosis (CAPA): A Cautionary Tale"

_ijms, 2022, doi:10.3390/ijms23063228_

Round 1

Reviewer 1 Report

Line 224, Line 227, Line 278: Please write the name of the species correctly A. fumigatus (italic and space after the dot)

Line 234: it is not Aspergillosis conidia, but Aspergillus conidia. Aspergillosis is the name of the disease and conidia was produced by the fungus Aspergillus.

Line 351: please write BDG completely since it is at the beginning of a sentence

Line 372: please write GAL completely since it is at the beginning of a sentence

Line 385-393: Please discuss more LFD and LFA, because these two kits can use easily in a simple laboratory? Which one is better in terms of sensitivity and specificity? etc.

Line 406-416:  Several studies have stated that although not as bad as amphotericin B, voriconazole has a nephrotoxicity effect. In COVID-19 patients, we know that renal injury often occurs because ACE2 which is a receptor for the virus is commonly found in the kidneys. It is worth discussing the effect of voriconazole administration on possible nephrotoxicity.

Author Response

Line 224, Line 227, Line 278: Please write the name of the species correctly A. fumigatus (italic and space after the dot)

RESPONSE: I have made the changes as advised.

Line 234: it is not Aspergillosis conidia, but Aspergillus conidia. Aspergillosis is the name of the disease and conidia was produced by the fungus Aspergillus.

RESPONSE: I have made the changes as advised. 

Line 351: please write BDG completely since it is at the beginning of a sentence

RESPONSE: I have made the changes as advised. 

Line 372: please write GAL completely since it is at the beginning of a sentence

RESPONSE: I have made the changes as advised. 

Line 385-393: Please discuss more LFD and LFA, because these two kits can use easily in a simple laboratory? Which one is better in terms of sensitivity and specificity? etc.

RESPONSE: I have discussed the different technologies introducing several new references which explore this topic. 

Line 406-416:  Several studies have stated that although not as bad as amphotericin B, voriconazole has a nephrotoxicity effect. In COVID-19 patients, we know that renal injury often occurs because ACE2 which is a receptor for the virus is commonly found in the kidneys. It is worth discussing the effect of voriconazole administration on possible nephrotoxicity.

RESPONSE: I have discussed the mechanism and risk factors for voriconazole induced nephrotoxicity . 

Reviewer 2 Report

The review by Worku highlights the immunopathologies of SARS-CoV2 and how it exacerbates covid associated pulmonary aspergillosis through dysregulation of local and systemic immunity. The review also focusses on the consequences of approved Covid therapies. The review provides knowledge about diagnosis of CAPA and possible treatments.

The manuscript is well written but have minor concerns. My feedback is outlined below.

  1. Unnecessary spaces throughout the manuscript.
  2. At some places, italics is used. line 206 page 6, line 246 page 7
  3. At some places, it is colored and bold. page 7, line 280.
  4. page 8 line 321 biochemical parameters...
  5. please abbreviate BDG initially
  6. Some mechanisms for pathology behind CAPA could be elaborated and discussed.
  7. Current diagnosis and treatment strategies in clinics should be discussed.
  8. Review states that there is problem in diagnosis and treatment for CAPA in Covid patients. this hypothesis needs to be clear in the beginning.
  9. the author should comment on CAPA in severe covid cases.
  10. Does CAPA occur in mild to moderate covid cases?
  11. How does CAPA occur in individuals with no underlying conditions and individuals with co-morbidities. can author elaborate this further?

Author Response

  1. Unnecessary spaces throughout the manuscript.- I have made the advised changes 
  2. At some places, italics is used. line 206 page 6, line 246 page 7- I have made the advised changes
  3. At some places, it is colored and bold. page 7, line 280. I have amended this as requested. 
  4. page 8 line 321 biochemical parameters...I have made the change as advised. 
  5. please abbreviate BDG initially- I have made the change as advised.
  6. Some mechanisms for pathology behind CAPA could be elaborated and discussed.- I have explored this, all changes are in red 
  7. Current diagnosis and treatment strategies in clinics should be discussed- I have explained that treatment strategies and how the diagnosis of CAPA is related to severe SARS-CoV2 infection and progression of disease .
  8. Review states that there is problem in diagnosis and treatment for CAPA in Covid patients. this hypothesis needs to be clear in the beginning. I have attended to this in the introduction 
  9. the author should comment on CAPA in severe covid cases. I have reinforced how CAPA is a disease found in the most severely unwell SARS-Cov2 patients occuring almost exclusively in ITU patients 
  10. Does CAPA occur in mild to moderate covid cases? Related to the above I have emphasised that CAPA is seen in severe disease and explained its presentation and how it contributes to the poor outcomes via necrosis, and infarction of the lung and subsequent invasion with multiorgan dysfunction
  11. How does CAPA occur in individuals with no underlying conditions and individuals with co-morbidities. can author elaborate this further?I have emphasised how severe COVID is a risk factor in of itself for CAPA and that the treatments used in the treatment of any hospitalised COVID can potentiate CAPA